# Relationship between weather conditions and the physicochemical characteristics of cladodes and mucilage from two cactus pear species

**Alba du Toit**[1]*, **Maryna de Wit**[1], **Hermanus J. Fouché**[2], **Sonja L. Venter**[3], **Arnold Hugo**[1]

**1** Department of Consumer Science, University of the Free State, Bloemfontein, Free State, South Africa, **2** Agricultural Research Council Animal Production, Irene, Gauteng, South Africa, **3** Agricultural Research Council Vegetable and Ornamental Plants, Pretoria, Gauteng, South Africa

☉ These authors contributed equally to this work.

* dutoita1@ufs.ac.za

**Data Availability Statement:** All files are available from Figshare database at https://doi.org/10.6084/m9.figshare.12514385.

## Abstract

Climate change, limited water resources and expected population increases would require crops which contribute toward more resilient, more productive, more sustainable and climate-smart food systems. The cactus pear is a drought-resistant and sustainable food source to humans and livestock alike. Cactus mucilage has multiple applications in the food and packaging industry. It is eco-friendly, economical, functional and has multiple health benefits. However, the researchers observed umpteen variations in extracted mucilage yield and viscosity every time the cladodes were harvested, making the standardisation of formulations troublesome. We aimed to examine the effect of weather conditions on the physicochemical characteristics of cactus pear cladodes and mucilage extracted over two seasons to understand these observed variations in mucilage characteristics. Forty cladodes, ten from each of *Opuntia ficus-indica* Algerian, Morado and Gymno-Carpo and *Opuntia robusta* Robusta were harvested every month from February to August in Bloemfontein, South Africa. Daily weather data were obtained, weight and moisture contents determined on cladodes and yield, viscosity, pH, conductivity and malic acid content determined on extracted mucilage. Pearson correlation coefficients were calculated between the weather conditions, cladode properties, and mucilage properties. Contrary to common belief, neither increasing cladode weight as they grow, nor rainfall were the leading causes of mucilage inconsistencies. However, the correlations showed a relationship between environmental temperatures, cladode pH and conductivity, and mucilage viscosity and yields. In hot summer weather, the pH was lower, which led to an abundance of positive ions in cladodes. The H+ ions neutralise the negative charges along the outstretched mucilage molecule, causing it to coil up, reducing the viscosity of the mucilage. Thus, environmental temperatures rather than rainfall or cladode maturity influenced the physicochemical characteristics of mucilage. The findings should make an essential contribution in predicting the physicochemical

**Funding:** This study was financially supported by a collaborative consortium between the Agricultural Research Council, Durban University of Technology and University of the Free State. The funds were rewarded to Dr M de Wit. The funders had no role in study design, data collection and analysis, decision to publish, or preparation of the manuscript.

**Competing interests:** The authors have declared that no competing interests exist.

characteristics of mucilage for specific food-related functions by observing the weather conditions.

## Introduction

The demand for agricultural products is expected to increase by 50% in 2030, to prevent hunger, as the global population is expected to increase [1]. However, climate change and limited water resources will impact agricultural productivity negatively, in fact, climate change will have the worst impact on crop productivity, and agricultural practices in countries already suffering high hunger levels [1]. Lobell et al. 2008 indicated that Southern Africa is one of the regions which will suffer most without sufficient adaptation measures. Adaptations in agriculture which contribute toward more resilient, more productive and more sustainable, climate-smart food systems will be necessary to establish food security in future [1,2]. The most viable adaptation option to increase food production and profits in the vulnerable hot and dry regions is switching to crops which are less impacted by climate change [2].

Cactus pears (*Opuntia* spp.) are increasingly recognised by researchers globally as a nutritious, drought-resistant, sustainable crop which could broaden the food base for livestock and humans alike. The plants are easily cultivated because of its ability to thrive during extreme heat, severe drought and in inferior soil [3,4]. Cactus pears will thrive in conditions of increased global warming as high temperatures, and an overabundance of $CO_2$ would increase cladode productivity and root growth [3].

Cactus pear cladodes are considered to be a multi-purpose crop. The consumption of young cladodes (nopalitos) as a fresh vegetable is spreading from Mexico to other parts of the world. Farmers, especially in Mexico and Brazil, already understand the economic, social and environmental advantages of using mature cactus pear cladodes as forage for livestock [5–7].

Mucilage, the slimy fluid which is abundant in the cladodes, is an eco-friendly, cheap, safe, nutrient-rich hydrocolloid with useful functional properties [8]. Mucilage consists mostly of indigestible, soluble fibre and contains minerals and antioxidants, which qualifies it as a low-calorie nutraceutical ingredient [9]. Cactus pear mucilage used to be regarded as waste but in recent years has become a trendy polymer, especially after its use in the packaging industry as a biodegradable, edible film and coating has been discovered [10,11]. Mucilage is now described as a valuable, added-value biomolecule material, widely available, economically profitable and a versatile polymer. In fact, mucilage is used in the food, pharmaceutical and cosmetic industries. Researchers have described mucilage as food stabilisers [12,13], thickeners, emulsifiers [14], a fat replacement agent [15], a stabilisingagent [16], a thickener [4], a suspension agent [17] and for encapsulations used in the pharmacological industry [18]. Thus, it possesses texture-modifying capabilities that improve or repair the textural characteristics of products [8,9,19,20]. The viscosity desired by the industry will vary according to the specific end-use. The food products themselves, the storage conditions and the preparation methods of food products have to be taken into consideration when using cactus pear mucilage [21]. Du Toit et al. 2019 used low viscosity mucilage for ice-cream and sorbet products to replace dairy or fats. In contrast, higher viscosity mucilage was preferred to replace egg or fats in mayonnaise formulations.

Cactus pears are an emerging as a crop which could provide a sustainable food source in hot and dry regions while providing mucilage as a functional product which could prove profitable as the demand is increasing worldwide. In a sustainable cactus pear orchard, the fruit

develops from spring to high summer. It is only after the fruit harvest that farmers need to diversify their income by harvesting cladodes for the extraction of mucilage. However, a problem with extracted mucilage is the inconsistency of its characteristics as the yield and viscosity constantly vary, making it difficult to standardise formulas and make predictions in terms of yields. It has been speculated that these differences occur as a result of hydration of cladodes as a consequence of the abundance of rain or extended periods of drought [22], and the maturity stages of cladodes have been named as an influencing factor on mucilage yields [23]. It is known that climatic conditions influence the quality of the fruit [24]. As part of a larger research project, cladodes were harvested over three years (2013–2015). In De Wit et al. 2019 we documented our findings from 42 cultivars harvested in 2013 in the dormant stage (winter) and proposed that the environmental conditions should be investigated, as the differences in mucilage yield and viscosity could be influenced by the weather. In 2014 we harvested eight cultivars over two growing seasons, namely the dormant stage (winter) and the post-harvest stage (summer). Again, we observed significant variations in mucilage yield and viscosity, not only between cultivars but also when cladodes from a single cultivar were harvested at different times of the year. So far, no studies focussed on explaining this phenomenon observed for many years of research on cactus pears growing in our orchard. We wanted to provide a possible explanation and open discussion amongst cactus pear mucilage researchers worldwide on this issue.

This paper examines the effect of weather conditions on the physicochemical characteristics of cactus pear mucilage extracted from mature cladodes harvested in 2015 over a six-month period. In doing so, the data from four cultivars and two species were pooled. We aim to correlate environmental temperatures and rainfall with the yield and viscosity of mucilage extracted over the six-month period that farmers would harvest cladodes in a dry land orchard. In particular, the relationships between temperature, rainfall, cladode size, cladode moisture content, mucilage acid content, mucilage conductivity, and mucilage viscosity will be investigated to address four research questions:

1. Does the size of cladodes influence the mucilage?

2. Does rainfall before harvest influence mucilage?

3. Do electrolytes influence mucilage?

4. Does the environmental temperature before harvest influence mucilage?

The findings should make an essential contribution in managing mucilage characteristics by using methods such as controlled environments or harvesting the cladodes at specific times or temperatures for specific purposes.

## Materials and methods

### Sample collection

Cactus pear cladodes were obtained from the Waterkloof experimental cactus pear orchard (GPS coordinates 29˚10'53" S, 25˚58'38" E) in the Free State, South Africa, located in the Bloemfontein district, 1 348 m above sea level. The site has an automatic weather service station (De Brug Weather Station) where the data for average and extreme maximum and minimum temperatures and rainfall were recorded daily from 2003 to 2017. The orchard had forty *Opuntia ficus-indica* cultivars and two *Opuntia robusta* cultivars laid out in a randomised complete block design (RCBD), with two replications for each cultivar and five plants per replication. After an in-depth selection process [25,26], three *O. ficus-indica* cultivars, namely Algerian, Morado and Gymno-Carpo, as well as *O. robusta* (Robusta), were used and the data

pooled in the current study. One cladode was harvested from each of the ten plants per cultivar for six months. Thus, for every month, the data from 40 samples were used to obtain the means for each month.

The harvesting of sample cladodes was done after the fruit harvest in high summer and before the next generation growth of cladodes in spring, between 9:00 and 11:15 on 25 February, 15 April, 20 May, 10 June, 15 July and 12 August 2015. In order to standardise the collection of cladodes, mature cladodes (cladodes that grew from spring 2014) were collected from the north side (maximum exposure to the sun in the Southern hemisphere) of the plant. The cladode had to be north/south orientated for maximum sun exposure, hip height (± 1m) and of good quality. The cladodes were systematically labelled, packaged and transported to the laboratory where they were individually weighed and refrigerated immediately.

A simple, economical, water and chemical-free, but effective process for the extraction of mucilage was developed and patented by the current authors [26]. The process involved cooking cladode pieces without the addition of water in a microwave oven for four minutes. The cooked cladode pieces were minced finely and centrifuged for 15 minutes at 8000 rpm. The supernatant native mucilage was decanted and forced through a 16 cm Kitchen Craft stainless steel sieve in order to separate the viscous mucilage liquid from remaining solid particles.

## Methods

The weight of each cladode sample (ten per cultivar), as well as extracted mucilage, was recorded using a Radwag PS 750/C/2 scale (g). The percentage yield of mucilage was calculated according to the original cladode weight.

$$\text{Yield of mucilage (\%)} = \frac{\text{extracted mucilage (g)}}{\text{cladode weight (g)}} \times 100$$

For moisture content determination [27], a standard circular sized segment was removed from each cladode and cut into smaller pieces by dividing the segment horizontally and then vertically into thirds.

For determination of mucilage viscosity, the line-spread test was performed using a sheet of paper, covered with glass, marked with concentric circles; each circle was 0.5 cm from the other, evenly measured off. The concentric circles were marked from one to 13 cm, indicating the distance from the central point. The circle was divided into eight parts and covered with a glass plate to provide an even and level surface. A one cm open-ended metal cylinder that corresponded with the smallest circle was placed on the chart and filled with five ml of mucilage. The cylinder containing mucilage was lifted for the mucilage to flow freely. When the mucilage stopped flowing, the distance it flowed was recorded (cm) on the eight lines dividing the circle. The distance values were added up to indicate the line-spread measurement value. The higher the reading of the line-spread test, the lower the viscosity of the mucilage as a result of it spreading further [28].

A calibrated Eutech pH 2700 pH/mV/˚C/˚F instrument was used to determine pH and conductivity (mV) at 22˚C. The tests were executed on mucilage each sample as soon as the extraction procedure was completed.

Samples were freeze-dried using a Perano freeze-drier for 72 hours at -60˚C for the determination of organic acids (chromatographic analysis). The dried powders were pre-treated by boiling 1 g at 80˚C for 15 min in 80% ethanol. For HPLC quantification, the samples were diluted 1:4, and centrifuged twice to remove all insoluble matter. The analysis was performed on a Thermo Surveyor HPLC with UV/Vis detection at 202 nm. The analytical column was a BioRad Aminex HPX 87H, the mobile phase 5 mM H2SO4, and the flow rate was 0.6 ml/min.

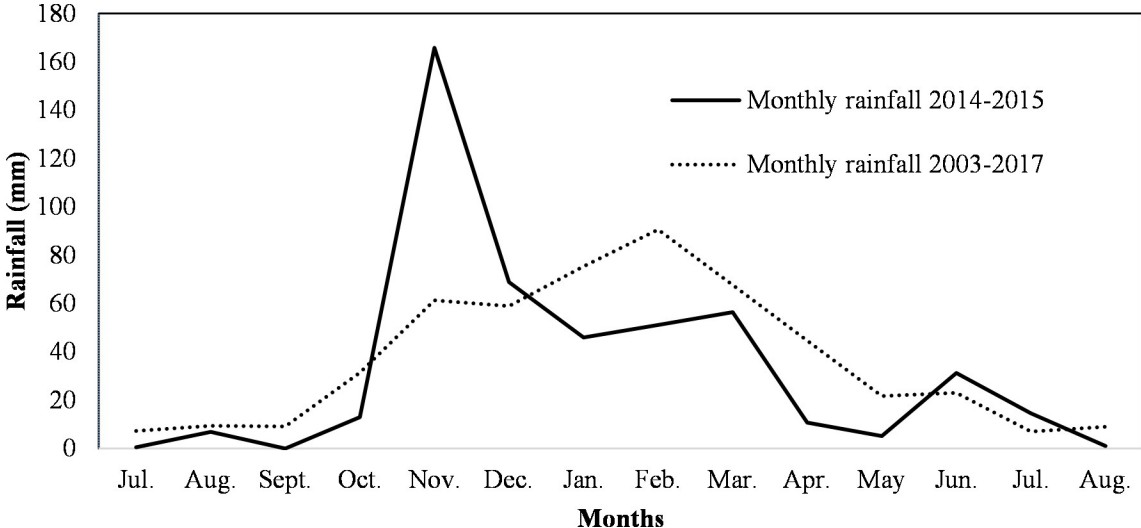

**Fig 1. Average total rainfall recorded for the cladode growth season from July to August 2003–2017 and 2014–2015 at Waterkloof farm, Bloemfontein, South Africa.**

## Statistical analysis

A one-way analysis of variance (ANOVA) procedure [29] was used to determine the effect of harvesting month on cladode and mucilage properties. The Tukey-Kramer multiple comparison test ($\alpha = 0.05$) was carried out to determine whether significant differences existed between treatment means [29]. Pearson correlation coefficients were calculated between environmental temperatures, rainfall and the cladode and mucilage properties [29].

## Results and discussion

The cladodes were harvested during 2015 from summer to late winter (February to August) developed on mother cladodes in spring (July 2014), and therefore the rainfall, minimum and maximum average and extreme temperatures for the period from July 2014 to August 2015 are shown in Figs 1–3.

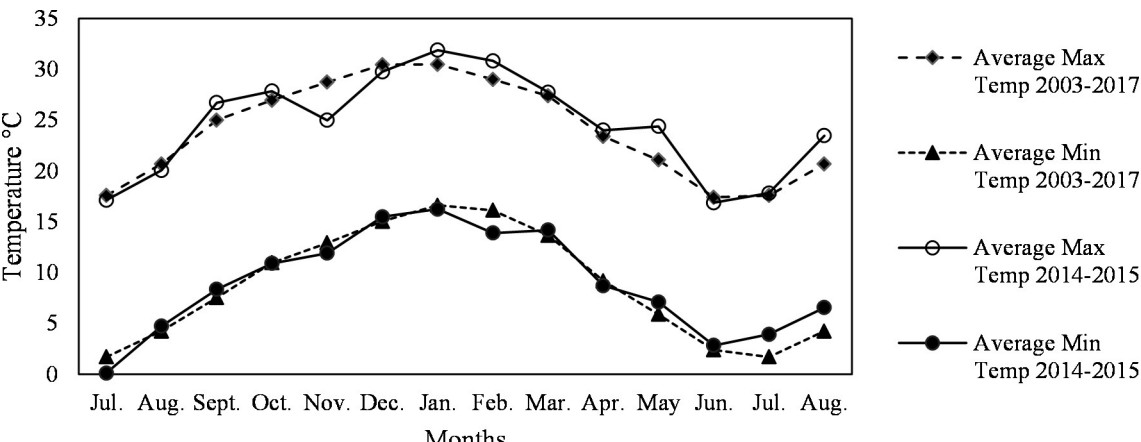

**Fig 2. Average minimum and maximum temperatures recorded for the cladode growth season from July to August 2003–2017 and 2014–2015 at Waterkloof farm, Bloemfontein, South Africa.**

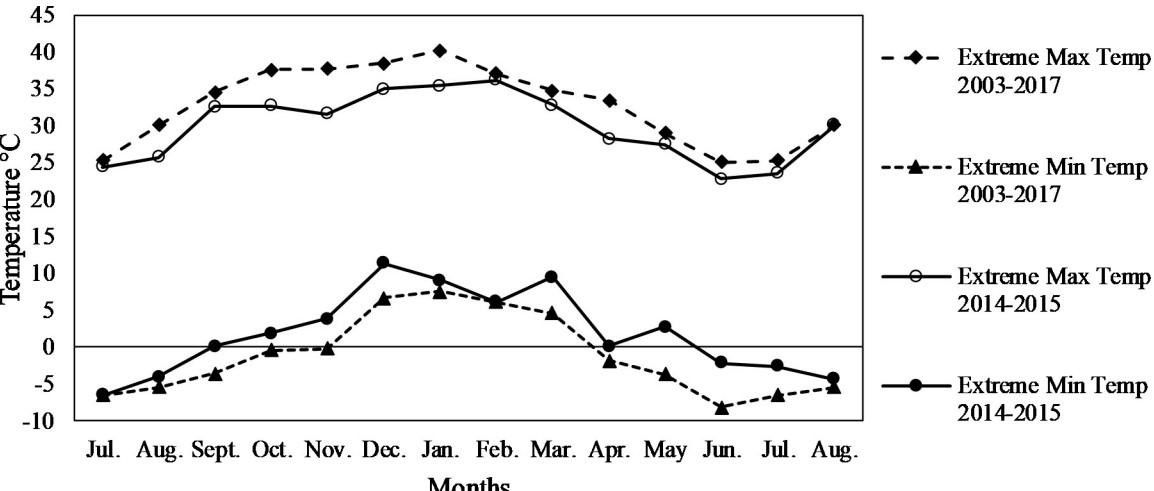

**Fig 3. Extreme minimum and maximum temperatures recorded for the cladode growth season from July August 2003–2017 and 2014–2015 at Waterkloof farm, Bloemfontein, South Africa.**

### Does cladode size influence mucilage?

In August, cladodes had the highest weight (880.55 g), while the lowest mucilage yield (27.60%) and highest viscosity (24.32 cm) was reported (Table 1). In February, cladodes had the lowest weight (599.30 g) while significantly high ($p < 0.001$) mucilage yield was reported in February (41.87%) and April (43.74%). In May (31.77%), June (30.94%), July (32.48%) and August (27.60%) mucilage yields were significantly lower ($p < 0.001$) than February and April while cladodes were steadily increasing in weight as they grew more mature.

De Wit et al., 2019 reported no linear relationship between cladode weight and mucilage yield ($r = -0.065$) in a comprehensive study of 42 cultivars harvested in the dormant stage and stated that cultivars with bigger and heavier cladodes were not necessarily selected for higher mucilage yields. It had always been assumed that bigger cladodes would produce higher mucilage yields. However, the assumption was disproven as cladode weight negatively correlated to the mucilage yield ($r = -0.4136$) (Table 2) and viscosity ($-0.4746$).

**Table 1. The weight and moisture content of cladodes harvested over six months (February to August 2015) and the yield, viscosity (Line-spread), pH, conductivity and malic acid content of mucilage extracted from cladodes over six months (February to August 2015).**

| | Cladodes | | Mucilage | | | | |
|---|---|---|---|---|---|---|---|
| **Harvest Month** | **Cladode Weight (g) n = 40** | **Moisture Content (%) n = 40** | **Yield (%) n = 40** | **Viscosity (cm) n = 40** | **pH n = 40** | **Conductivity (mS/cm) n = 40** | **Malic acid content (g/ L) n = 4** |
| **February** | 599.30±199.39[a] | 91.15±1.52 | 41.87±12.78[b] | 31.61±4.58[b] | 3.97±0.07[a] | 166.84±4.33[e] | 3.68±0.83[b] |
| **April** | 745.13±201.45[abc] | 91.09±1.88 | 43.74±7.98[b] | 30.34±2.93[b] | 4.03 ±0.07[ab] | 164.46±5.53[de] | 3.24±0.46[b] |
| **May** | 613.10±53.23[a] | 89.83±1.46 | 31.77±11.21[a] | 26.60±4.33[a] | 4.12±0.06[b] | 157.43±4.27[d] | 2.73±0.40[b] |
| **June** | 677.60±88.79[ab] | 89.10±2.22 | 30.94±11.33[a] | 25.45±3.26[a] | 4.82±0.33[c] | 117.86±23.63[c] | 2.64±0.42[b] |
| **July** | 843.18±120.80[bc] | 89.05±3.42 | 32.48±14.36[a] | 27.22±4.40[a] | 5.18±0.08[d] | 91.79±7.76[b] | 2.70±0.21[b] |
| **August** | 880.55±200.09[c] | 90.72±1.39 | 27.60±10.17[a] | 24.32±4.59[a] | 5.42±0.08[e] | 81.52±5.61[a] | 0.69±1.24[a] |
| **Significance level** | ($p < 0.001$) | ($p = 0.575$) | ($p < 0.001$) | ($p < 0.001$) | ($p < 0.001$) | ($p < 0.001$) | ($p < 0.001$) |
| **Means ± SD** | 726.48±134.58 | 90.15±1.26 | 34.73±10.95 | 27.59±3.94 | 4.59±0.07 | 129.98±4.83 | 2.62±0.31 |

Month means with different superscripts in the same column differ significantly

**Table 2. Pearson correlation coefficients between rainfall and maximum temperatures and the weight and moisture content of cladodes harvested over six months (February to August 2015) and the yield, viscosity (Line-spread), pH, conductivity and malic acid content of mucilage extracted from cladodes over six months (February to August 2015).**

| | Cladode Weight (g) | Cladode Moisture content (%) | Mucilage Yield (%) | Mucilage Viscosity (cm) | Mucilage pH | Mucilage Conductivity (mS/cm) | Mucilage Malic acid (g/L) | Monthly Rainfall (mm/month) | Maximum Temperature (°C) |
|---|---|---|---|---|---|---|---|---|---|
| **Cladode Weight (g)** | 1 | -0.1119$^{NS}$ | -0.4136$^{NS}$ | -0.4746$^{NS}$ | 0.8235$^{*}$ | -0.8256$^{*}$ | -0.7055$^{NS}$ | -0.5900$^{NS}$ | -0.4582$^{NS}$ |
| **Cladode Moisture content (%)** | | 1 | 0.6066$^{NS}$ | 0.5667$^{NS}$ | -0.4648$^{NS}$ | 0.4793$^{NS}$ | 0.0800$^{NS}$ | 0.1167$^{NS}$ | 0.8500$^{*}$ |
| **Mucilage Yield (%)** | | | 1 | 0.9634$^{**}$ | -0.7872$^{*}$ | 0.7835$^{*}$ | 0.7896$^{*}$ | 0.4679$^{NS}$ | 0.5601$^{NS}$ |
| **Mucilage Viscosity (cm)** | | | | 1 | -0.7789$^{*}$ | 0.7681$^{*}$ | 0.8324$^{*}$ | 0.5762$^{NS}$ | 0.6451$^{NS}$ |
| **Mucilage pH** | | | | | 1 | -0.9991$^{***}$ | -0.8022$^{*}$ | -0.3912$^{NS}$ | -0.6294$^{NS}$ |
| **Mucilage Conductivity (mS/cm)** | | | | | | 1 | 0.7870$^{*}$ | 0.3915$^{NS}$ | 0.6319$^{NS}$ |
| **Mucilage Malic acid (g/L)** | | | | | | | 1 | 0.6395$^{NS}$ | 0.2923$^{NS}$ |
| **Monthly Rainfall (mm/month)** | | | | | | | | 1 | 0.3405$^{NS}$ |
| **Maximum Temperature (°C)** | | | | | | | | | 1 |

$^{NS}$ = Not Significant

$^{*}$ = $p < 0.05$

$^{**}$ = $p < 0.01$

$^{***}$ = $p < 0.001$

### Does rainfall before harvest influence mucilage?

As Bloemfontein lies within a summer rainfall area, it is common for very little rain to fall during the colder months. The cumulative rainfall was 445.4 mm for July 2014 to June 2015 and 407.84 mm from July 2015 to June 2016 (Fig 1). The average yearly rainfall recorded over fifteen years (2003 to 2017) was higher (512 mm). The average total rainfall recorded for the orchard from 2003 to 2017 as well as the monthly rainfall from July 2014 to August 2015 is seen in Fig 1. Dry months were recorded for July to Oct 2014 (usually the dry season) and from January to May 2015 (usually the rainy season). July 2014 (0.5 mm) was the driest month and November 2014 (165.85 mm) the wettest month. In November and December 2014, more than the average monthly rainfall was recorded.

It had always been assumed that cladodes would have higher moisture content when water (rainfall) is plentiful. However, the moisture content in cladodes did not increase or decrease significantly between February (91.15%) and August (90.72%). Although not significant, the lowest cladode moisture content was recorded in June (89%) (Table 1). There was only a slight decrease in cladode moisture content (2.05%) during the winter months when less than 25 mm of rain was reported (April to August) compared to the summer months (January to March) when more than 45 mm of rain fell. Thus, the moisture content in cladodes harvested over the six months did not differ significantly even though the rainfall fluctuated markedly between summer (high rainfall) and winter months (low rainfall) (Fig 1). In this study, the correlation between moisture content and monthly rainfall (r = 0.117) was not significant (p > 0.05), which showed that there was no discernible relationship between rainfall and cladode moisture content. In fact, monthly rainfall did not correlate strongly with any of the

cladode or mucilage properties (Table 2). Results from this study agrees with De Wit et al. (2019), who stated that the moisture content of cladodes was not correlated with the viscosity or yield of mucilage.

## Do electrolytes in cladodes influence mucilage?

Mucilage molecules are hetero-polysaccharides, consisting of different monosaccharides with varying uronic acid content (both a carbonyl and a carboxylic acid), and high molecular weight ($4.3 \times 10^6$ g/mol) [30]. The molecules are negatively charged, unbranched, long-chain polymers that repel themselves and each other, causing them to stretch out and distribute in a solute, resulting in an increase in its viscosity [30,31]. The viscosity of the mucilage solution is strongly dependent on the ion concentration of the solution [30,31]. The influence of pH and ionic strength on the viscosity of mucilage solutions after extraction was described by Du Toit et al. (2019); the mucilage solutions showed an increase in viscosity in the alkaline region and a decrease in the acidic region [21,31,32] and a decrease in viscosity with an increase in electrolyte concentration [21].

The electrolytes present in cladodes fluctuates hourly in cladodes because of the special type of carbon fixation used by cactus pear plants. Crassulacean acid metabolism (CAM) is the photosynthesis pathway that allows cactus pear cladodes to retain water along with obtaining $CO_2$. It causes the stomata to open at night for the fixation of $CO_2$ when water-loss would be limited while malic acid is constructed and accumulates in the cladodes [33]. This accumulation of malic acid causes the pH to drop towards dawn while its deconstruction during the hours of sunlight causes the pH of cladodes to rise [33].

The pH of mucilage increased consistently over the months of harvest from summer to late winter. It was significantly ($p < 0.001$) lower in February (3.97) to May (4.12), thereafter it increased significantly ($p < 0.001$) in the colder weather of June (4.82) and increased again in July (5.18) and August (5.42) (Table 1). At the same time, the malic acid content decreased (not significantly) steadily from February (average 3.68 g/L) to July (2.70 g/L) and dropped significantly ($p < 0.001$) in August (0.69 g/L). The conductivity of mucilage decreased significantly ($p < 0.001$) from February (166.84 mS/cm) to May (157.43 mS/cm) and continued to decrease significantly ($p < 0.001$) to June (117.86 mS/cm), July (91.79 mS/cm) and August (81.52 mS/cm) (Table 1).

A significant ($p < 0.05$) strong negative correlation (r = -0.8) between higher cladode malic acid content and lower mucilage pH was observed. The pH values showed an almost perfect and highly significant ($p < 0.001$) negative relationship (r = -0.999) to ion concentration. Thus, there were higher concentrations of free ions in mucilage when the pH was lower. In terms of viscosity, mucilage extracted from cladodes harvested in February (31.61 cm) and April (30.34 cm) had significantly lower viscosity ($p < 0.001$) than mucilage extracted from cladodes harvested in May (26.60 cm), June (25.45 cm), July (27.22 cm) and August (24.32 cm) (Table 1). Thus, the more acidic mucilage had lower viscosity (Table 1). This was indicated by a strong and significant ($p < 0.05$) negative correlation (r = -0.78) between pH and low viscosity, a positive and significant ($p < 0.05$) correlation between conductivity and low viscosity (r = 0.77) and a significant ($p < 0.05$) positive relationship between low viscosity and high malic acid content (r = 0.832). A negative correlation (r = -0.63, $p > 0.05$) between pH and maximum temperatures indicated that warmer temperatures correlated moderately with lower pH values. The positive (r = 0.96) and significant ($p < 0.01$) relationship between high mucilage yield and low viscosity (Table 2) indicated that lower viscosity mucilage produces higher yields. Furthermore, a strong and significant ($p < 0.05$) negative correlation between mucilage yield and pH (-0.79), and strong significant ($p < 0.05$) positive correlation between yield and

conductivity (r = 0.78) as well as yield and malic acid content (r = 0.79) indicated a strong correlation between mucilage acidity and higher yields (Table 2). A similarly strong positive relationship between mucilage yield and viscosity (r = 0.69) was also found in the study on 42 cultivars [25].

## Does the environmental temperature before harvest influence mucilage?

It was reported [34] that continuous high (40°C) and low (4°C) temperatures caused severe stress to young cactus pear cladodes, but in this case, the cactus pear plants grew in an orchard, and are subjected to ordinary day-night and seasonal weather which allow for acclimation and temperature adaption to occur [35]. In fact, it was found that in CAM plants, the optimal $CO_2$ fixation rates adapt to current growth temperatures. Thus, CAM plants growing at higher temperatures, have optimal $CO_2$ fixation rates at higher temperatures and vice versa [35].

In cold weather, the mucilage in the cladodes plays a role in the tolerance to intracellular freeze dehydration and provides noncolligative protection to cell membranes [36,37]. The accumulation of low molecular substances in mucilage, such as sugars and proteins serve as cryoprotectants as it restricts the mobility of intracellular water, however, below -6°C, when the extracellular ice crystals draw water from mucilage and out of the plant cells, permanent cell damage occurs [37,38]. Thus, most cactus pear plants are extremely vulnerable to frost.

However, cactus pear plants are extremely tolerant to high temperatures up to 65°C and can survive for 60 min at 70°C, especially the older cladodes [38]. Its tolerance to high temperatures is enhanced when acclimation is gradual. Therefore, hot weather is not a limiting factor for the cultivation of cactus pears as a crop [38].

Moreover, the $CO_2$ metabolism of cactus pear cladodes is most effective when nocturnal (minimum) temperatures are between 10 and 20°C (optimal 14°C) [3,38]. The temperature of the cladode itself may be lower than the air temperature at night, because of cooling, which takes place as a result of transpiration and heat loss by infrared radiation [38].

For the fifteen years of recorded temperatures (2003–2017), the average maximum temperatures remained above 25°C while the average minimum temperatures remained above 10°C (favourable cladode growth temperatures) from October to March (Fig 2). Comparable favourable conditions for cladode development prevailed during the 2014–2015 growth season (Fig 2). For the colder months, between April and September, the average maximum temperatures for 2003 to 2017, as well as June 2014 to Aug 2015 remained above 15°C, while the average minimum temperatures remained above 0°C (Fig 2).

Extreme minimum temperatures (Fig 3) during 2014–2015 were recorded in June (-2.13°C), July (-2.55°C) and August (-4.32°C) of 2015 (Fig 3). Over the 15 years from 2003 to 2017 (Fig 3), extreme minimum temperatures only dropped below -6°C (causing cell damage) three times in July 2006 (-6.2°C), June 2014 (-8.18°C) and July 2014 (-6.5°C).

The average maximum temperatures were positively correlated to cladode moisture content (r = 0.85, p > 0.05) (Table 2). The average maximum temperatures also showed a positive correlation to low mucilage viscosity (r = 0.64, not significant) and moisture content showed a moderate positive correlation with low mucilage viscosity (r = 0.57) and high mucilage yield (r = 0.67) (not significant). In Du Toit et al. (2019), the viscosity of extracted mucilage in solutions decreased when heated up and increased when cooled down. Thus, in hot weather, the moisture content of cladodes was higher, the mucilage yields higher, and the viscosity lower.

## Conclusion

The demand for cactus pear mucilage is increasing worldwide; thus, it is necessary to understand and predict its physicochemical characteristics in order to produce a profitable

functional product. In this study, it was found that neither cladode weight nor rainfall content was relevant to the moisture content of cladodes or the viscosity or yield of mucilage. However, the abundance of electrolytes which occurred in warmer weather (lower mucilage pH and higher conductivity) had a strong correlation to higher yields and lower viscosity. In fact, warmer weather conditions were positively correlated to higher cladode moisture content, lower mucilage viscosity and higher mucilage yields.

The viscosity of mucilage was lower in warm summer months, because of the abundance of positive ions in cladodes when the pH was lower in hot weather. Thus, in hot weather, the higher $H^+$ ions neutralised the negative charges that were open along the mucilage molecule, causing a reduced repulsion and extension of mucilage molecules. Accordingly, a physical change in the molecular shape and configuration occurs as it coils up, which reduce the viscosity of the mucilage, as fewer water molecules would be bound along the molecule. Mucilage of lower viscosity was more readily separated from the cladode solids during extraction; thus, the mucilage yield was higher during warmer months.

The environmental temperatures rather than rainfall or the size of cladodes influenced the physicochemical characteristics of mucilage. A relationship between temperature during harvest, mucilage pH, conductivity, viscosity, and yield were established in this study. Thus, lower viscosity and higher yields of mucilage were obtained from cladodes harvested in hot and dry conditions due to the physiological changes which altered the shape of the mucilage molecules.

Further studies are needed to further resolve issues on cactus pear and mucilage characteristics. Studies are underway on the effect of high temperatures and fertilisation on cactus pear plants. Further research is proposed to measure the chlorophyll fluorescence to assess the photosynthetic energy conversion and a greenhouse rainfall exclusion experiment.

Cactus pears have high commercial potential for cultivation in hot regions to produce mucilage, as high environmental temperatures may increase yields. Cactus pears offer new opportunities to farmers in harsh and dry regions as a sustainable, drought-resistant and multi-purpose crop.

## Author Contributions

**Conceptualization:** Alba du Toit.

**Data curation:** Arnold Hugo.

**Formal analysis:** Arnold Hugo.

**Funding acquisition:** Maryna de Wit, Sonja L. Venter.

**Investigation:** Alba du Toit.

**Methodology:** Alba du Toit.

**Project administration:** Alba du Toit, Maryna de Wit.

**Resources:** Hermanus J. Fouché.

**Supervision:** Maryna de Wit, Sonja L. Venter, Arnold Hugo.

**Validation:** Hermanus J. Fouché.

**Visualization:** Alba du Toit.

**Writing – original draft:** Alba du Toit.

**Writing – review & editing:** Maryna de Wit, Sonja L. Venter, Arnold Hugo.

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
