## [Decision Letter · Decision Letter 0]

9 Jun 2020

PONE-D-20-00751

Relationship between weather conditions and the physicochemical characteristics of cactus pear (Opuntia spp.) cladodes and mucilage

PLOS ONE

Dear Dr. Toit,

Thank you for submitting your manuscript to PLOS ONE. After careful consideration, we feel that it has merit but does not fully meet PLOS ONE’s publication criteria as it currently stands. Therefore, we invite you to submit a revised version of the manuscript that addresses the points raised during the review process.

We look forward to receiving your revised manuscript.

Kind regards,

Zhenhai Han, PhD

Academic Editor

PLOS ONE

Journal Requirements:

Reviewers' comments:

Reviewer's Responses to Questions

**Comments to the Author**

1. Is the manuscript technically sound, and do the data support the conclusions?

Reviewer #1: Partly

Reviewer #2: Partly

Reviewer #3: No

2. Has the statistical analysis been performed appropriately and rigorously? 

Reviewer #1: Yes

Reviewer #2: Yes

Reviewer #3: No

3. Have the authors made all data underlying the findings in their manuscript fully available?

Reviewer #1: Yes

Reviewer #2: No

Reviewer #3: Yes

4. Is the manuscript presented in an intelligible fashion and written in standard English?

Reviewer #1: Yes

Reviewer #2: Yes

Reviewer #3: Yes

5. Review Comments to the Author

Reviewer #1: The manuscript studied the effect of weather conditions on the physicochemical characteristics of cactus pear cladodes and mucilage extracted over two seasons to understand the observed variations in mucilage characteristics. Daily weather data were obtained, weight and moisture contents determined on cladodes and yield, viscosity, pH, conductivity and malic acid content determined on extracted mucilage. They found that correlations showed a relationship between environmental temperatures, cladode pH and conductivity, and mucilage viscosity and yields.

Below are the comments for you to consider.

Major points:

1. In MM, “Algerian, Morado and Gymno-Carpo, as well as O. robusta (Robusta), were used in the current study”. Did the author find any differences between the cultivars? I did not see individual data of the cultivars in the tables.

2. What is the aim of study? What kind of viscosity of mucilage is preferred in food industry?

3. Was the data repeated for at least two years?

4. What factor caused the changed pH value during different months?

Reviewer #2: REVIEW

Manuscript PONE-D-20-00751: "Relationship between weather conditions and the physicochemical characteristics of cactus pear (Opuntia spp.) cladodes and mucilage" by Alba Du Toit et al.

1. a) This study presents the results of original research. Analyzes the changes in cladodes of three cultivars of Opuntia ficus-indica and one of Opuntia robusta. This article states in the title: “characteristics of cactus pear (Opuntia spp.)” but the genus Opuntia has 150+ species. The abbreviation "spp." (plural) indicates "several species". Although correct for two species, it could be misleading to readers assuming several species. I suggest rephrasing it to “Relationship between weather conditions and the physicochemical characteristics of cladodes and mucilage of two prickly cactus species” or mention three cultivars of Opuntia ficus-indica and one of O. robusta.

b) The article is well-written and informative. However, although the sample size per cultivar and month seems adequate, the period of study does not cover one phenological year. It is restricted to months from the peak of summertime to winter. It is unfortunate not having data for springtime, particularly to get more variance in the studied attributes and weather factors. Also, there is no chance of comparison among years to assist in data interpretation. c) The study is correlational and offers little insight into the causal factors related to mucilage content of cladodes. I would expect some degree of simple experimentation like subjecting plants to different irrigation or to high temperature schedules to help clarify the role of weather variables on cladode attributes and mucilage yield.

Regarding the correlations, are not size and ontogenetic stage confounded variables? i. e. There is a correlation between cladode age and yield? If so, mucilage percentage is related to ontogeny with cladodes harvested in summer still accumulating biomass related to support and mucilage remaining constant. Hence, yield (%) could be related to the increase in structural biomass.

d) The other cladode variables, pH, conductivity and malic acid content, are not related to phenological events—i. e. varying photosynthetic activity through the seasons? Interestingly, rainfall seems well correlated (although marginally not significant given the small sample size). These issues could have been easily solved by measuring chlorophyll fluorescence to assess photosynthetic energy conversion and doing a greenhouse rainfall exclusion experiment.

e) The poor correlation of cladode attributes with rainfall probably derives from the lagged plant response to environmental variables. For temperature, the lag is usually more rapid than for water uptake. Also, there is almost no variance in cladode moisture content, suggesting full turgor most of the time. This measurement again highlights the relevance of a longer period of study. Bloemfontein, South Africa has a relatively mild, mesic climate, much gentler than the one where O. ficus-indica and especially O. robusta grow. Local climate could explain, in part, the lack of correlation with rainfall, even considering lags.

f) The presentation of results is unusual by mixing discussion and results. It is not a major issue, but is uncommon. I prefer the more standard separation between the Results section without references, and the Discussion section where the authors contrast the results with the literature.

g) Figures 1-3 can be grouped into one. They all use the same X-axis and refer to weather variables.

As far as I know, the results reported have not been published elsewhere, but similar results can be gleaned from a search in academic databases.

2. The experiments, statistics, and other analyses are performed satisfactorily. The number of samples is not stated in the tables, but the reader assumes is n=10. I recommend adding the number of observations. They are described in sufficient detail.

3. The conclusions are properly presented but as I remarked above, there is no Discussion section.

4. The article is well-written and is presented in an intelligible fashion and is written in standard English.

5. I see that the research meets all applicable standards for the ethics of experimentation and research integrity.

6. The article adheres to appropriate reporting guidelines and community standards for data availability. However, it is limited in its scope and fails in fulfilling the expectations.

Reviewer #3: The manuscript "Relationship between weather conditions and the physicochemical characteristics of cactus pear (Opuntia spp.) cladodes and mucilage" in interesting, but the experimental design is not adequate because it was not rigorously conducted. The experiment has only two replications per each species or cultivar, which is not enough for statistical analyses. In addition, a one-way ANOVA procedure was used to determine the effect of harvesting month on cladode and mucilage properties. What about the species or cultivar in the statistical analysis? Species or cultivars can differ in physicochemical characteristics, but it appears that authors mixed data from all species. In addition, Pearson correlation coefficients were calculated between environmental temperatures, rainfall and the cladode and mucilage properties. Again, what about the corrrelations among environmental temperatures, rainfall and the cladode and mucilage properties for each species or cultivar?

6. PLOS authors have the option to publish the peer review history of their article (what does this mean?). If published, this will include your full peer review and any attached files.

Reviewer #1: No

Reviewer #2: No

Reviewer #3: No

---

## [Author Response · Author response to Decision Letter 0]

7 Jul 2020

Response to reviewers

Reviewer 1

1. In MM, “Algerian, Morado and Gymno-Carpo, as well as O. robusta (Robusta), were used in the current study”. Did the author find any differences between the cultivars? I did not see individual data of the cultivars in the tables. Our recently published articles focused on the statistical differences between the nutritional composition, functional, and rheological aspects of the four different cactus pears over the six-month period. However, in this publication, we pooled data to increase the sample size to prove our point. 

2. What is the aim of study? What kind of viscosity of mucilage is preferred in food industry?

 Mucilage from cactus pears is attracting more and more attention in the scientific community with the umpteen newly discovered uses such as the production of biodegradable plastics and in the food industry. The viscosity desired by the industry will vary according to the specific end-use. An example of the use of lower or higher viscosity mucilage was added to the manuscript (line 79-85). We wanted to provide the necessary knowledge to manage mucilage characteristics by using methods such as controlled environments or harvesting the cladodes at specific times or temperatures for specific purposes.

3. Was the data repeated for at least two years? Thank you for pointing out that this point was not clearly described in the manuscript. The manuscript was edited to explain the work we did over three years (lines 95-110). In De Wit (2019), we documented our findings from 42 cultivars harvested in the dormant stage (winter) in 2013. In 2014 we harvested eight cultivars over two growing seasons, namely the dormant stage (winter) and the post-harvest stage (summer). Thus, in this article, we only show the data for one year, but similar effects were observed consistently in all three years. In 2015 we harvested over six months and pooled the data to be able to analyse the data in more detail and thoroughly over the entire period from after the fruit harvest in summer to the dormant stage in winter. 

We have been working with mucilage over many years (du Toit and de Wit registered a patent in 2011 on the extraction of mucilage). We believe that the biggest problem with mucilage is the inconsistencies in the characteristics not only between cultivars but at different times of harvests in the same cultivar. 

We believe that the findings should make an essential contribution to understanding and managing mucilage.

4. What factor caused the changed pH value during different months? It is known that the cactus pear is a CAM plant. CAM causes pH changes between day and night. We found that changes in pH occurred as a result of temperature changes between summer and winter months. We explained that pH is one of the factors that influence mucilage viscosity. The abundance of electrolytes which occurred in warmer weather (lower mucilage pH and higher conductivity) had a strong correlation to higher yields and lower viscosity. In fact, warmer weather conditions were positively correlated to higher cladode moisture content, lower mucilage viscosity and higher mucilage yields.

With this paper, we wanted to open the discussion and encourage further research amongst the international cactus pear research community by publishing these findings. 

Reviewer 2

This article states in the title: “characteristics of cactus pear (Opuntia spp.)” but the genus Opuntia has 150+ species. The abbreviation "spp." (plural) indicates "several species". Although correct for two species, it could be misleading to readers assuming several species. I suggest rephrasing it to “Relationship between weather conditions and the physicochemical characteristics of cladodes and mucilage of two prickly cactus species” or mention three cultivars of Opuntia ficus-indica and one of O. robusta. The title was changed, thank you for pointing it out and for the useful suggestion. We omitted the word “prickly pear” as it is not an internationally recognized term.

b) The article is well-written and informative. However, although the sample size per cultivar and month seems adequate, the period of study does not cover one phenological year. It is restricted to months from the peak of summertime to winter. It is unfortunate not having data for springtime, particularly to get more variance in the studied attributes and weather factors. Also, there is no chance of comparison among years to assist in data interpretation. Thank you for pointing out that the description in the manuscript was unclear. Sentences were added to clarify it (lines 87-91 and 132-133). We were restricted to the sustainable and practical use of the cactus pear orchard, i.e. allowing the fruit to develop in spring until mid-summer. It is only from after the fruit harvest that farmers want to diversify their income and harvest cladodes for the extraction of mucilage. 

c) The study is correlational and offers little insight into the causal factors related to mucilage content of cladodes. I would expect some degree of simple experimentation like subjecting plants to different irrigation or to high temperature schedules to help clarify the role of weather variables on cladode attributes and mucilage yield.

 Thank you for the suggestion. We agree that the experimentation, suggested by the reviewer, would give insight into the causal factors related to the mucilage content. We are currently doing studies on the effect of high temperatures as well as the effect of fertilization on different cactus pear cultivars. Currently, an experiment on the effect of nitrogen fertilization on mucilage properties is undergoing. A sentence was added as a recommendation for further study. It is this kind of research and discussions that we wanted to highlight and open amongst the international cactus pear research community by publishing these findings. We believe that not only the international cactus pear research community but also farmers and the food industry would benefit from the information as presented.

Regarding the correlations, are not size and ontogenetic stage confounded variables? Size and development stage are part of the development of the plants growing in an orchard under normal conditions; we are merely sharing our findings with the international cactus pear community in order to show that temperature and not rainfall and cladode size influence the characteristics of mucilage.

e. There is a correlation between cladode age and yield? If so, mucilage percentage is related to ontogeny with cladodes harvested in summer still accumulating biomass related to support and mucilage remaining constant. Hence, yield (%) could be related to the increase in structural biomass. We agree that it could be assumed that the mucilage may well stay constant. With the specific extraction method, we used, more mucilage was extracted when the viscosity was low. Thus, the total amount of mucilage was not determined, but the amount that was possible to be extracted from the solids.

The relationship between cladode weight and mucilage yield was discussed under the first research question in the article. The conclusion was that the cladode weight (as the structural biomass increased over the months) was not correlated to the mucilage viscosity or yield.

d) The other cladode variables, pH, conductivity and malic acid content, are not related to phenological events—i. e. varying photosynthetic activity through the seasons? Interestingly, rainfall seems well correlated (although marginally not significant given the small sample size). These issues could have been easily solved by measuring chlorophyll fluorescence to assess photosynthetic energy conversion and doing a greenhouse rainfall exclusion experiment. Thank you very much for your suggestion. At present, we are limited to obtaining samples in a dry land orchard where the plants are subjected to typical weather and seasonal changes. We are looking into collaborating with researchers to do further experimentation in controlled greenhouse environments. The suggestion for further studies was added as a recommendation to the manuscript.

e) The poor correlation of cladode attributes with rainfall probably derives from the lagged plant response to environmental variables. For temperature, the lag is usually more rapid than for water uptake. Also, there is almost no variance in cladode moisture content, suggesting full turgor most of the time. This measurement again highlights the relevance of a longer period of study. Bloemfontein, South Africa has a relatively mild, mesic climate, much gentler than the one where O. ficus-indica and especially O. robusta grow. Local climate could explain, in part, the lack of correlation with rainfall, even considering lags. Thank you for your suggestion in doing further research. With the publication of these findings, we would like to encourage further research and for data to be available from other parts of the world and climatic conditions in order to compare weather conditions to mucilage characteristics. 

f) The presentation of results is unusual by mixing discussion and results. It is not a major issue, but is uncommon. I prefer the more standard separation between the Results section without references, and the Discussion section where the authors contrast the results with the literature. We believe that this method was the most appropriate to incorporate the data, discussions, and literature in order to answer each research question. 

g) Figures 1-3 can be grouped into one. They all use the same X-axis and refer to weather variables.

As far as I know, the results reported have not been published elsewhere, but similar results can be gleaned from a search in academic databases. We agree that it would be possible to group all the data in one figure, but we believe that the number of figures was well within the allowed amount of figures and tables and that the data is more apparent in separate figures.

2. The experiments, statistics, and other analyses are performed satisfactorily. The number of samples is not stated in the tables, but the reader assumes is n=10. I recommend adding the number of observations. They are described in sufficient detail. Thank you for pointing out this oversight. We will include the number of observations in the tables. The data in tables was from ten samples of four cactus pears, thus 40 samples per month in total.

3. The conclusions are properly presented but as I remarked above, there is no Discussion section. Thank you, we appreciate the positive feedback.

4. The article is well-written and is presented in an intelligible fashion and is written in standard English. Thank you, we appreciate the positive feedback.

5. I see that the research meets all applicable standards for the ethics of experimentation and research integrity. Thank you, we appreciate the positive feedback.

6. The article adheres to appropriate reporting guidelines and community standards for data availability. 

However, it is limited in its scope and fails in fulfilling the expectations. Thank you, we appreciate the positive feedback. 

This paper had a specific focus which was to reach a consensus in the cactus pear research community that environmental temperature and not rainfall or cladode size cause the inconsistencies in mucilage characteristics. We believe that we fulfilled our intension. 

Reviewer 3

The experimental design is not adequate because it was not rigorously conducted. The experiment has only two replications per each species or cultivar, which is not enough for statistical analyses. We believe that we used an adequate number of samples for statistical analysis. The orchard was laid out with two replications for each cultivar and five plants per replication. Three O. ficus-indica cultivars, namely Algerian, Morado and Gymno-Carpo, as well as O. robusta (Robusta), were used in the current study. One cladode was harvested from each of the ten plants from two replications from four cultivars over six months. Thus, for every month, the data from 40 individually studied samples were used to obtain the means for each month represented in the tables. We added sentences in the manuscript to clarify this point (lines 131-133).

In addition, a one-way ANOVA procedure was used to determine the effect of harvesting month on cladode and mucilage properties. What about the species or cultivar in the statistical analysis? Species or cultivars can differ in physicochemical characteristics, but it appears that authors mixed data from all species. Our recently published articles focused on the statistical differences between the nutritional composition, functional, and rheological aspects of the four different cactus pears over the six-month period. However, in this study, we pooled the data to determine the influence of environmental temperature and rainfall on the characteristics in mucilage of different cultivars and species. 

In addition, Pearson correlation coefficients were calculated between environmental temperatures, rainfall and the cladode and mucilage properties. Again, what about the corrrelations among environmental temperatures, rainfall and the cladode and mucilage properties for each species or cultivar?

 Thank you for your suggestion in doing further analysis of the data. We are currently working on such a study. We observed significant variations in mucilage yield and viscosity, not only between cultivars but also when cladodes from a single cultivar that were harvested at different times of the year. So far, no studies focussed on explaining this phenomenon. With this article, we would like to establish a consensus in the research community that environmental temperature and not rainfall alone cause the inconsistencies in mucilage characteristics.

---

## [Decision Letter · Decision Letter 1]

29 Jul 2020

Relationship between weather conditions and the physicochemical characteristics of cladodes and mucilage from two cactus pear species

PONE-D-20-00751R1

Dear Dr. Toit,

We’re pleased to inform you that your manuscript has been judged scientifically suitable for publication and will be formally accepted for publication once it meets all outstanding technical requirements.

Kind regards,

Zhenhai Han, PhD

Academic Editor

PLOS ONE

Additional Editor Comments (optional):

Reviewers' comments:

Reviewer's Responses to Questions

**Comments to the Author**

1. If the authors have adequately addressed your comments raised in a previous round of review and you feel that this manuscript is now acceptable for publication, you may indicate that here to bypass the “Comments to the Author” section, enter your conflict of interest statement in the “Confidential to Editor” section, and submit your "Accept" recommendation.

Reviewer #1: All comments have been addressed

2. Is the manuscript technically sound, and do the data support the conclusions?

Reviewer #1: Yes

3. Has the statistical analysis been performed appropriately and rigorously? 

Reviewer #1: Yes

4. Have the authors made all data underlying the findings in their manuscript fully available?

Reviewer #1: Yes

5. Is the manuscript presented in an intelligible fashion and written in standard English?

Reviewer #1: Yes

6. Review Comments to the Author

Reviewer #1: The MS studied relationship between weather conditions and the physicochemical characteristics of

cladodes and mucilage from two cactus pear species. The comments are properly revised.

7. PLOS authors have the option to publish the peer review history of their article (what does this mean?). If published, this will include your full peer review and any attached files.

Reviewer #1: No

---

## [Editor Report · Acceptance letter]

3 Aug 2020

PONE-D-20-00751R1 

Relationship between weather conditions and the physicochemical characteristics of cladodes and mucilage from two cactus pear species 

Dear Dr. du Toit:

I'm pleased to inform you that your manuscript has been deemed suitable for publication in PLOS ONE. Congratulations! Your manuscript is now with our production department. 

Kind regards, 

on behalf of

Dr. Zhenhai Han 

Academic Editor

PLOS ONE